# Insight into the protein solubility driving forces with neural attention

**Daniele Raimondi**[1], **Gabriele Orlando**[2], **Piero Fariselli**[3], **Yves Moreau**[1] *

**1** ESAT-STADIUS, KU Leuven, Leuven, Belgium, **2** SWITCH Lab, KU Leuven, Leuven, Belgium, **3** Università di Torino, Torino, Italy

* yves.moreau@kuleuven.be

**Data Availability Statement:** All relevant data are publicly available from the respective papers. The code described in this study is available at: https://bitbucket.org/eddiewrc/skade/src.

## Abstract

Protein solubility is a key aspect for many biotechnological, biomedical and industrial processes, such as the production of active proteins and antibodies. In addition, understanding the molecular determinants of the solubility of proteins may be crucial to shed light on the molecular mechanisms of diseases caused by aggregation processes such as amyloidosis. Here we present SKADE, a novel Neural Network protein solubility predictor and we show how it can provide novel insight into the protein solubility mechanisms, thanks to its neural attention architecture. First, we show that SKADE positively compares with state of the art tools while using just the protein sequence as input. Then, thanks to the neural attention mechanism, we use SKADE to investigate the patterns learned during training and we analyse its decision process. We use this peculiarity to show that, while the attention profiles do not correlate with obvious sequence aspects such as biophysical properties of the aminoacids, they suggest that N- and C-termini are the most relevant regions for solubility prediction and are predictive for complex emergent properties such as aggregation-prone regions involved in beta-amyloidosis and contact density. Moreover, SKADE is able to identify mutations that increase or decrease the overall solubility of the protein, allowing it to be used to perform large scale in-silico mutagenesis of proteins in order to maximize their solubility.

## Author summary

The solubility of proteins is a crucial biophysical aspect when it comes to understanding many human diseases and to improve the industrial processes for protein production. Due to its relevance, computational methods have been devised in order to study and possibly optimize the solubility of proteins. In this work we apply a deep-learning technique, called *neural attention* to predict protein solubility while "opening" the model itself to interpretability, even though Machine Learning models are usually considered *black boxes*. Thank to the attention mechanism, we show that i) our model implicitly learns complex patterns related to emergent, protein folding-related, aspects such as to recognize *β*-amyloidosis regions and that ii) the N-and C-termini are the regions with the highes signal fro solubility prediction. When it comes to enhancing the solubility of proteins, we, for the first time, propose to investigate the synergistic effects of tandem mutations instead

**Funding:** DR is founded by a FWO post-doctoral fellowship. The funders had no role in study design, data collection and analysis, decision to publish, or preparation of the manuscript.

**Competing interests:** The authors have declared that no competing interests exist.

of "single" mutations, suggesting that this could minimize the number of required proposed mutations.

## Introduction

Proteins have evolved within the different cellular environments to improve or preserve their functions, maintaining at the same time a degree of hydrophobicity necessary to proper fold, and enough solubility to prevent protein precipitation. Solubility is then a fundamental ingredient that must be properly balanced to maintain the protein functions and not aggregating [1]. Protein solubility is also an essential aspect in diagnostic and therapeutic applications [2, 3], as well as being a critical requirement for protein homeostasis [1, 4, 5]. Solubility deficit can hamper protein-based drug development, generating insoluble protein precipitates that can be toxic and may elicit an immune response in patients [6, 7]. Protein aggregation is considered an hallmark of more than forty human diseases, that span from neurodegenerative illness, to cancer and to metabolic disorders such as diabetes [8].

Since the soluble expression of proteins is crucial for protein production [9] for both pharmacological and research goals [10], in-silico bioinformatics models have been developed to predict i) the solubility of proteins [10–13] or ii) the solubility change upon mutation [9, 12, 14, 15], in order to, respectively, i) select the most likely soluble proteins [12] and ii) run in-silico mutagenesis to increase the solubility of existent proteins [9].

The methods developed so far in both categories use various Machine Learning (ML) approaches, such as Neural Networks (NN) [10], Gradient Boosting Machines [11], Support Vector Machines (SVM) [13], Logistic Regression [12] or simpler statistical methods [9, 14]. Most of these approaches compute predictions starting from just the proteins sequence, with the exception of CamSol, which uses PDB structures [14]. Among the sequence-based features used, the most common are: i) biophysical propensity scales values (e.g. hydrophobicity, charge), ii) various forms of k-mers frequencies (such as mono, bi- or tri-peptides occurrences), iii) predictions from other methods (e.g. disorder, Secondary Structure, Relative Solvent Accessibility or aggregation predictors) and iv) *global* features such as sequence length and the fraction of residues exposed to the solvent.

A common issue that the methods predicting the solubility of proteins had to face is the fact that the input protein sequences may have very different lengths, and indeed building ML models able to work with protein sequences is a common task in structural bioinformatics. From the ML standpoint, this task is not trivial because the variable length of proteins poses some issues to conventional ML methods, such SVM or Random Forests. This problem is usually addressed by using sliding window techniques to predict each residue independently [16, 17], but different solutions are needed when a single prediction must be associated to an entire protein sequence [13, 14, 18], since the information content of an entire sequence needs to be *shrunk* into a single predictive scalar value.

Neural Networks (NN) are flexible models that can elegantly address this issue. The classical approaches consist in building a pyramid-like architecture [10] that takes the protein sequence as input and reduces it to a fixed size through subsequent abstraction layers, ending with a feed-forward sub-network that yields the final scalar prediction.

Here we propose a novel solution to this issue, which has been inspired by the neural attention mechanisms developed for Natural Language Processing and machine translation [19, 20]. Our model is called SKADE and uses a neural attention-like architecture to elegantly process the information contained in protein sequences towards the prediction of their solubility.

By comparing it with state of the art methods we show that it has competitive performances while requiring as inputs just the protein sequence.

Additionally, the use of neural attention allows our model to be *interpreted*, showing that the learned patterns correlate with complex sequence aspects such as the presence of aggregating regions or the protein contact density, while it does not correlate with more trivial aspects such as biophysical propensity scales or solvent accessibility.

We also show that, even if it has not been specifically trained for the task, SKADE can distinguish between mutations that increase or decrease the proteins' solubility. This, coupled with the fact that it can generate hundreds of predictions per second, makes it ideal to perform in-silico optimization of protein solubility. To show its potential in this sense, we performed a complete in-silico mutagenesis of the UPF0235 protein MTH_637, computing both the effect of all the possible single mutations and all possible pairs of *tandem* mutations ($> 2 \times 10^6$ pairs). This allowed us to investigate the possible effects of *synergistic* interactions between mutations, indicating that, in certain regions of the proteins, the implementation of pairs of mutations could have a larger effect that the sum of the effects of independent mutations. Finally, we show that the predicted synergistic effects have a significant correlation with the average contact distances between residues, extracted from the protein PDB structure, suggesting that SKADE is able to catch a glimpse of complex emergent properties such as the contact density.

## Materials and methods

### Datasets

To train and test our model, we used the protein solubility datasets adopted in [10, 11]. Using the same training/testing data and procedure allowed us to compare the performances of SKADE with the most recently published methods. The training set contains 28972 soluble and 40448 insoluble proteins that have been annotated with the pepcDB [21] "soluble" (or subsequent stages) annotations in [12]. The test dataset contains 1000 soluble and 1001 insoluble proteins, and has been compiled by [22].

To validate the ability of SKADE to distinguish between variants increasing or decreasing the overall solubility of the protein, we adopted the dataset used in CamSol [14], which contains 142 variants known to increase or decrease the solubility of 42 proteins.

### Neural attention for protein sequences

SKADE is a NN model which consists in two sub-networks: the predictor network *P* and the attention network *A*.

The NN takes as input just a the target proteins sequences, without using evolutionary information or other kinds of annotations. As a first step, each residue in the input sequence is translated into a trainable 20-dimensional embedding encoding the 20 possible amino acids. The embedded sequences are then sorted and padded, in order to maximize the efficiency of the NN computations on GPUs.

Both *A* and *P* are constituted by 2 layers of bidirectional Gated Recurrent Unit (GRU) [23] networks with 20 hidden dimensions. The GRU network runs on the input sequences, outputting 20 dimensional vector for each residue. Since the GRU is bidirectional, for each protein it provides two $20 \times L$ tensors as output, where L is the length of the sequence. The two tensors obtained from the forward and backward pass of the bidirectional GRU are concatenated into a $40 \times L$ and further processed by a linear layer that produces a single scalar value for each residue, obtaining a $1 \times L$ tensor for each protein.

At this point the $A$ and $P$ network differ: the last layer of the predictor $P$ has a LeakyReLU activation while the attention network A has a SoftMax activation that is globally applied over the entire $1 \times L$ tensor T, ensuring that $\sum_{i=0}^{L} T_i = 1$.

To obtain the final prediction, for each protein the scalar product $a^T p$ between the $1 \times L$ tensors $a$ and $p$, respectively obtained from the $P$ and $A$ sub-networks is computed, allowing the SoftMax-ed attention vector $a$ to *select* the position on the vector $p$ that should be given highest relevance for the final prediction. The resulting value is passed through a Sigmoid activation, constraining the final output of SKADE withing the 0-1 range.

Conceptually, the predictor network P should assign different solubility-related values to each residue in the protein, considering also the local sequence context. In parallel, the attention network $A$ should learn on which protein regions its attention should be focused.

The final model has 25462 trainable parameters, uses a batch size of 1001 with an initial learning rate of 0.01. We used the Adam optimizer with a L2 regularization equal to $10^{-6}$ and it is trained for 50 epochs. The model has been implemented in Pytorch version 1.0.1 [24]. The code is freely available at https://bitbucket.org/eddiewrc/skade/src.

## Validation procedure and performance evaluation

In order to compare our results with [10, 11], we reproduced their validation procedure. We trained our model on the 69420 proteins in the trainset and we tested it on the 2001 proteins in the test set from [22]. We used the widely adopted Sensitivity (Sen), Specificity (Spe), Precision (Pre), Accuracy (Acc), Matthews Correlation Coefficient (MCC), Area Under the ROC curve AUC and Area Under the Precision Recall Curve (AUPRC) metrices to evaluate the predictions. The Balanced Accuracy (BAC) is computed as the arithmetic mean of Sen and Spe.

To assess the ability of SKADE to distinguish between variants increasing or decreasing the solubility of proteins, we followed the procedure used in SODA [9].

## Results

### SKADE predicts protein solubility from just the protein amino acid sequence

The most recently developed tools for the protein solubility prediction [10, 11] use various kinds of features. These can be divided into sequence related features (e.g. k-mers and their frequencies of occurrence, biophysical propensity scales sequence descriptions), global features (e.g. sequence length, molecular weight, fraction of residues with certain biophysical properties) and structural features, such as secondary structure assignments (SS) and Relative Solvent Accessibility (RSA) [10, 11].

During the development of SKADE we chose to use only the protein sequences as inputs because the addition of any other kind of features would pose certain restrictions to our model. For example, in [25] it has been shown that, although extremely valuable, the use of Multiple Sequence Alignments (MSAs) as features could bias predictors towards more studied proteins (e.g. proteins for which many homologous sequences are known). Another recent study we conducted [26] showed that the use of biophysical propensity scales coupled with sophisticate ML method could be just a discrete and intrinsically limited way to find an optimized embedding description of the amino acids in the proteins, and using entirely random scales could give extremely similar results [26]. Finally, our rationale to avoid the use of global features is that, since the solubility of a protein is a crucial step for its expression and large scale manufacture, we envision SKADE as a tool for the in-silico optimization of protein solubility, and thus we deemed global features such as the protein length, the molecular weight and

**Table 1. Table showing the comparison between SKADE and the state of the art tools benchmarked in [22].**

| Methods | Sen | Spe | Pre | Acc | MCC | AUC | AUPRC |
|---|---|---|---|---|---|---|---|
| SKADE | 66 | **81** | **78** | 73 | 47 | **82** | **82** |
| PaRSnIP | **76** | 72 | 70 | **74** | **48** | 82 | 80 |
| DeepSol S1 | 75 | 71 | 69 | 73 | 46 | 81 | 81 |
| PROSO II | 67 | 68 | 69 | 64 | 34 | 74 | 71 |
| CCSOL | 54 | 54 | 51 | 54 | 8 | - | - |
| SOLPRO | 62 | 58 | 51 | 60 | 20 | - | - |
| PROSO | 58 | 57 | 54 | 58 | 16 | - | - |
| RPSP | 52 | 51 | 44 | 52 | 3 | - | - |
| SCM | 65 | 57 | 42 | 60 | 21 | - | - |

the absolute charge as not suitable for our framework, since they are not characteristics on which we want to act directly while perform the optimization.

S6 Fig shows a PCA of the learned embeddings, and S2 Text contains their actual 20-dimensional values.

## SKADE positively compares with state of the art predictors

We trained and tested SKADE following the same procedure adopted in [10, 11], which are two of the most recently developed solubility predictors. This allowed us to compare SKADE with the most relevant state of the art methods, extending the benchmark initially proposed in [22]. This procedure involved a training set containing 69420 proteins and a test set of 2001 proteins. The results obtained by SKADE and the comparison with other methods is shown in Table 1.

From DeepSol [10] we reported the performance of their S1 model, which is the DeepSol version that uses only protein sequence information. PaRSnIP and all the other method, on the other hand, use any kind of features, some of which include information related to Secondary Structures, Relative Solvent Accessibility and other biophysical aspects. Notwithstanding this limitation, Table 1 shows that SKADE has the best AUPRC and the best AUC, on par with PaRSnIP.

## The role of attention

As shown in Fig 1, SKADE's NN is divided into two sub-networks. The sub-network A is used to compute the per-residues SoftMax-ed attention values $a$ while the sub-network P is used to compute the per-residue predictions $p$. The final solubility prediction for each protein is obtained as $\sigma(a^T p)$, where $\sigma$ is a Sigmoid activation. This value is thus a linear combination between $p$ and $a$, and the $A$ networks is responsible to assign the coefficients that are used to weight the per-residues predictions proposed by $P$. This is analogous to a bias-less Logistic Regression in which the weights are dynamically *predicted* for each sample instead of being learned once for all, and it allows SKADE to work with input sequences with arbitrary length, making it suitable to bioinformatics applications.

Moreover, since $\sigma(a^T p) = \sigma(\sum_i^L a_i p_i)$, with $\sigma$ monotonically increasing, residues $i$ with a positive value of $a_i \times p_i$ steer the prediction towards the class 1 (soluble), while residues with negative values are actually "voting" for the class 0. This means that these per-residue values can be considered as the *solubility profile* of the protein (see S3–S5 Figs for the attention and prediction profiles of the proteins Q8TC59, Q9HBE1 and P25984).

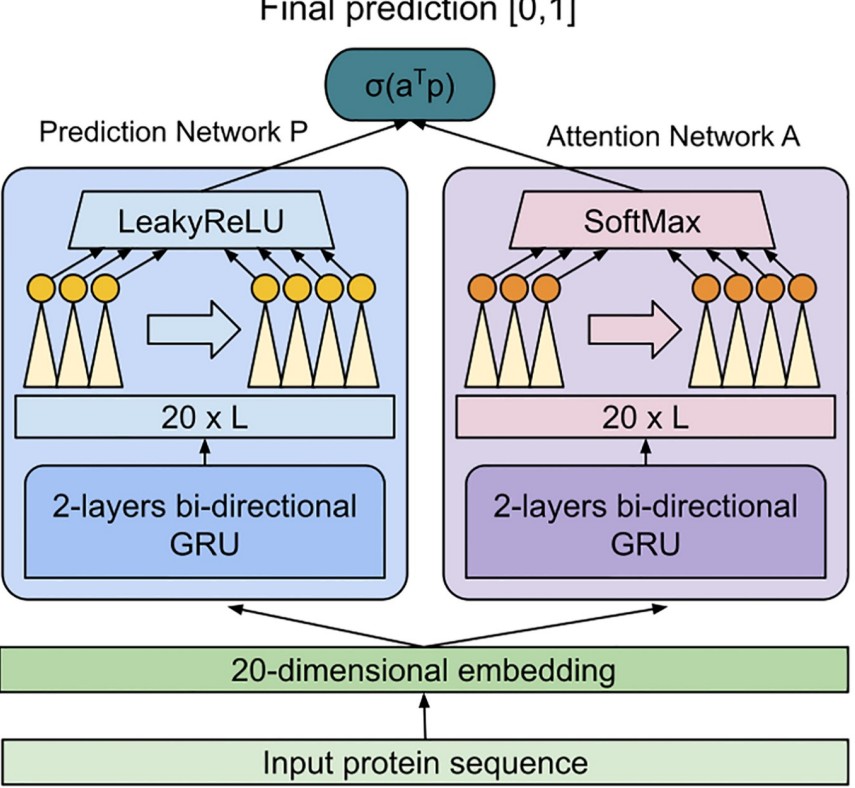

**Fig 1. Figure showing the structure of the NN.** The protein sequence is translated into a 20 dimensional embedding and then passed to the predictor network P (left) and the attention network A (right). Each of these subnetworks contain 2 layers of bi-directional GRU network, followed by a feed-forward. The predictor ends with a LeakyReLU activation, while the attention network has a SoftMax activation. Finally, the $1 \times L$ outputs of the two networks are reduced to a predictive, probability-like, scalar value by a dot product operation, followed by a SIgmoid activation.

In order to analyse the behavior of SKADE and to investigate the learned patterns of attention, we tried to correlate the per-residue attention and solubility profiles to known biophysical properties, such as hydrophobicity, polarity, charge, volume and the propensities of the residues being in $\alpha$-helical or $\beta$-sheet conformation, but no correlation with these values has been detected (see Table A1 in S1 Text), indicating that the A and P networks behavior cannot be directly associated with trivial sequence properties. We also tested the correlation of the profiles against in-house Relative Solvent Accessibility (RSA) predictions, obtaining a Pearson correlation of $r = 0.0237$.

## SKADE solubility profiles detect aggregation patches

We then tested the predictive power of SKADE's solubility profiles on the AMYL dataset [27], which contains 34 amyloidogenic proteins whose aggregating regions have been used in [28] to benchmark the performances of per-residue in-silico aggregation predictors. The AMYL dataset contains annotations indicating the involvement of each residue in $\beta$-amyloidosis aggregation, for a total of 7732 residues, 2599 of which are responsible for aggregation.

SKADE's solubility profiles show an interesting signal towards the prediction of aggregating regions in these proteins. The profiles have an AUC of 65 even though SKADE has not been trained on the proteins in the AMYL dataset nor on the $\beta$-amyloidosis aggregation task at any

**Table 2. Comparison of the performance of SKADE with state of the art aggregation predictors on the amyl33 dataset [27].** Results are reported from [28].

| Method | Sen | Spe | BAC | MCC |
|---|---|---|---|---|
| AgMata | 43 | 84 | 66 | 25 |
| PASTA 2 (85 specificity) | 41 | 85 | 63 | 24 |
| PASTA 2 (90 specificity) | 30 | 90 | 60 | 22 |
| AMYLPRED2 | 39 | 84 | 62 | 22 |
| **SKADE** | 55 | 71 | 63 | 20 |
| MetAmyl | 52 | 71 | 62 | 17 |
| Tango | 14 | 96 | 55 | 14 |
| Aggrescan | 35 | 79 | 57 | 13 |
| FishAmyloid | 14 | 94 | 54 | 10 |
| FoldAmyloid | 21 | 87 | 54 | 8 |

point. To better contextualize the magnitude of this signal, in Table 2 we compared the performance of SKADE's solubility profiles with state of the art tools that have been specifically developed to predict aggregation. We can see that even though the solubility profiles of SKADE are a byproduct of the neural attention model and have been obtained in a completely unsupervised way with respect to the task at hand, SKADE provides quite competitive performance when it comes to identifying aggregating regions in proteins.

We can thus conclude that, although the neural attention profiles do not show correlation with trivial biophysical aspects of the protein sequence, the $a_i \times p_i$ per-residue solubility profiles that are responsible for the final prediction show an interesting correlation with a complex and still only partially understood behavior such as $\beta$-amyloidosis aggregation of proteins.

## The N- and C-termini are the most relevant regions for the protein solubility prediction

Further analysing the data extracted from the neural attention used in SKADE, in Fig 2, we show the mean and median values of attention $a$ (red) and prediction $p$ (blue) in function of the relative position of the residues in the sequence. S1 Fig shows the distributions of the $a_i \times p_i$ per-residues solubility profile value, where $i$ is the residue position in each sequence. From both figures it clearly appears that SKADE focuses its attention to the N- and C-termini of the proteins, indicating that they contain signal for the prediction of the protein solubility. To investigate whether this behavior is an artifact of the recurrent modules in our NN, we tested SKADE on the 2001 sequences in the test set, but first removing the initial and final 20% of residues from each of them. The obtained AUC is very close to random (0.55), indicating that the beginning and the end of the sequences might indeed carry an important signal for the prediction of the protein solubility. On the contrary, when we test SKADE on the 2001 sequences in the test set after removing the central residues (located in the relative sequence positions between the 20th and 80th percentiles), the performances are very similar to normal (0.81 of AUC).

To further ensure that this behavior is due to the signal carried by the N- and C-termini regions and not an artifact of SKADE's architecture, we repeated the same experiments by using the DeepSol S1 webserver to predict the 2001 proteins in the test set after removing i) the initial and final 20% of residues from each sequence, and ii) the central portion of each protein (from the 20th to the 80th percentile). Similarly to the results obtained with SKADE, DeepSolS1 produces almost random predictions when the N- and C-termini are removed from the input sequences (AUC = 0.53), and exactly normal predictions when the central part

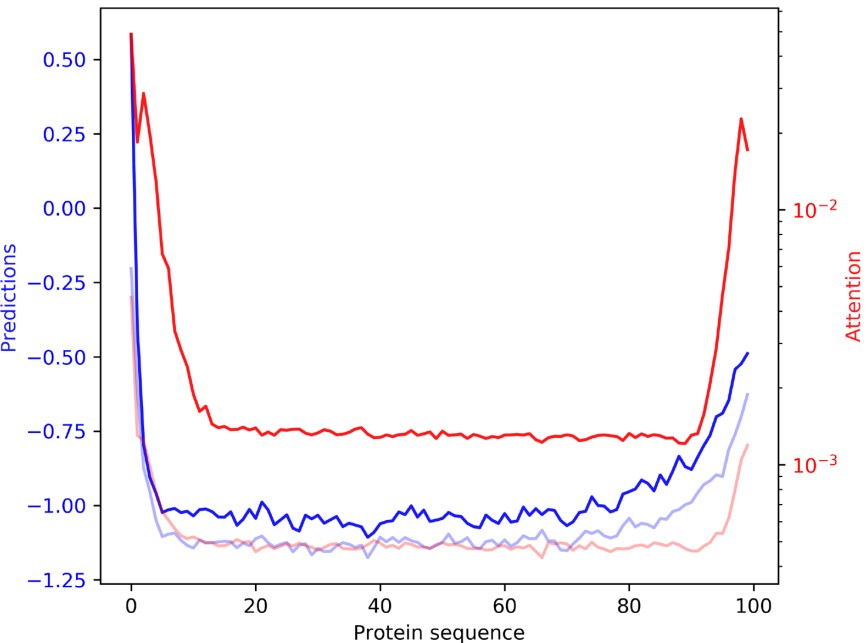

**Fig 2. Plot showing the mean (solid red) and median (light red) attention values, compared with the mean (solid blue) and median (light blue) prediction values, in function of the position in the sequence.** On average, SKADE assigns higher values to positions close to the N- and C-termini.

of the protein is removed (AUC = 0.81). This shows that also DeepSol S1, which is based on a completely different architecture from SKADE, implicitly exploits the information contained in the extremities of the sequences to perform its prediction.

We also searched in literature whether the relevance of N- and C-termini was already established, and although we did not find exhaustive studies analysing this behavior, a number of papers investigated specific cases. For example, it has been shown that mutating the N-terminus of the mitochondrial aminoacyl-tRNA synthetases [29], sperm whale myoglobin [30] and hen egg-white lysozyme [31] enhances their expression and solubility.

### Unsupervised prediction of solubility change upon mutation

One of the possible applications of protein solubility predictors is the in-silico optimization of protein sequences to enhance their solubility, for example by selecting the smallest possible set of variants able to increase the solubility of the sequence while minimizing the structural and functional alterations to the original protein.

To do so, it is necessary that the prediction methods are able to distinguish between variants that increase or decrease the overall solubility of the sequence. Unfortunately, very little experimental data is available in this sense, and in particular, to the best of our knowledge, no data concerning the effect of multiple mutations or insertions/deletions on the solubility of proteins is available.

To evaluate the ability of SKADE to identify mutations increasing or decreasing protein solubility, we used the CamSol [14] dataset, which contains 56 experimentally validated mutations. S7 Fig shows the distribution of the variants in CamSol on the corresponding protein sequences.

Since SKADE is designed to take the whole target protein sequence as input, to evaluate the solubility change upon mutation we predicted both the wildtype and the mutated target

**Table 3. Table showing the comparison of the unsupervised predictions of SKADE on the CamSol dataset of mutations.** Results have been preported form [9].

| Methods | Accuracy | Total |
|---|---|---|
| SODA | 100.0 | 56/56 |
| CamSol | 96 | 54/56 |
| SKADE | 71 | 40/56 |
| SolPro | 71 | 40/56 |
| PROSO II | 57 | 32/56 |

sequence, computing the difference in predicted solubility between the wildtype ($WT_s$) and the mutant ($MUT_s$) as $\Delta S = MUT_s - WT_s$.

In Table 3 we used the CamSol dataset to compare the performance of SKADE with methods designed for the task of predicting the effect of mutations on the protein solubility. To do so we reproduced the benchmark performed in SODA [9], where 4 existing predictors have been tested. SKADE correctly identifies 69% of the mutations increasing solubility and 100% of the mutations decreasing it, with an AUC of 82 and an AUPRC of 99%.

An important consideration is that while methods such as SODA (see Table 3) have been specifically designed and trained to discriminate mutations increasing or decreasing the solubility, SKADE faces this task in a completely unsupervised way, since it has been trained to predict the solubility of entire protein sequences.

## In-silico mutational screening and analysis of the synergistic effects of mutations

SKADE is a NN and thus it can be easily run in parallel on GPUs, computing predictions for hundreds or thousands of sequences per second. This can be used to compute in-silico mutational screening of proteins, as shown in Fig 3 for UPF0235 protein MTH_637 (Uniprot ID: O26734), which is present in the test dataset [22]. The protein is annotated as soluble in the dataset, and SKADE predicts it correctly (score of 0.733). We implemented each possible mutation in O26734 and we computed the the change in solubility $\Delta S = MUT_s - WT_s$, meaning that positive $\Delta S$ (shown in red) indicate that the mutation increases the solubility of the protein, while negative $\Delta S$ values (in blue) decrease it. Fig 3 shows that, since the predicted solubility of O26734 is already very high, most of the mutations have the effect of decreasing the predicted solubility. In particular, there are few regions with very high effect, such as the residues from 1 to 25, from 49 to 51 and close to the C-terminal. Among the possible mutations, it appears that most of the mutations of a wildtype residue into a Met are predicted to heavily decrease the solubility. We listed the mutations with the larges $\Delta S$ in S1 Text.

An aspect that it is not usually considered when performing classical in-silico mutational screenings is the difference between the effects of two mutations $m_i$ and $m_j$ when they are considered independently or in combination (i.e. both implemented at the same time), thus investigating the possible *synergistic* effects of mutations. This screening of pairs of mutations is usually not doable due to the extremely high number of possibilities that needs to be tested, but SKADE is fast enough to allow also the exhaustive analysis of the effects combinations of variants. As an example, we ran this experiment on the protein O26734, which is 103 residues long, and we tested the solubility changes $\Delta S_{i,j}$ due to all the possible pairs of variants $m_i$ and $m_j$, for a total of $(103 \times 20 \times 102 \times 20)/2 = 2101200$ mutations. We then computed the change in solubility due to two independent mutations $m_i$ and $m_j$ from Fig 3 as $\Delta S_{single} = \Delta S_i + \Delta S_j$, and the change in solubility obtained when both $m_i$ and $m_j$ are implemented in the sequence

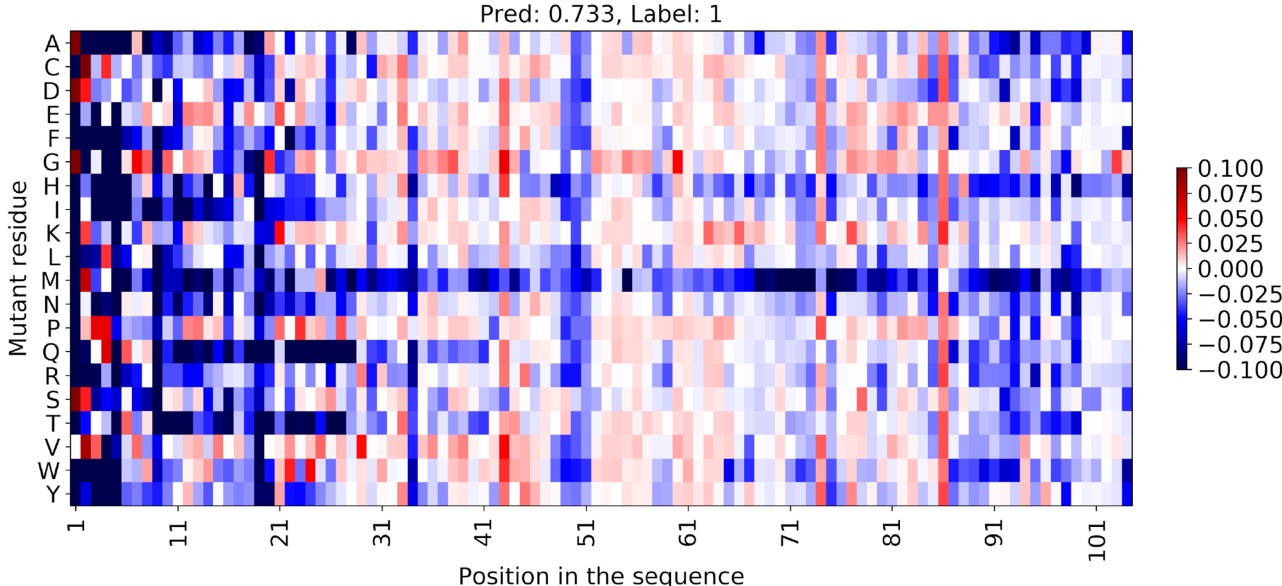

**Fig 3. In-silico mutational screening of UPF0235 protein MTH_637 (O26734), showing the effect on the solubility of the protein of every possible mutation.** The change in solubility is computed as $\Delta S = MUT_s - WT_s$, meaning that positive $\Delta S$ (shown in red) indicate that the mutation increases the solubility of the protein, while negative $\Delta S$ values (in blue) decrease it.

$\Delta S_{pair} = \Delta S_{i,j} = (MUT_{i,j} - WT)$. We then computed the synergistic versus individual effects as $\Delta E = \Delta S_{single} - \Delta S_{pair}$ for each pair of variants $m_i$ and $m_j$.

In Fig 4 we averaged the $\Delta E$ for each position of O26734, indicating on which pairs of sequence positions the synergistic effects have a stronger effect than two independent variants, averaged over all the possible mutations that could be implemented in those positions. Fig 4 shows that SKADE's model predicts that most of the positions in O26734 are more likely affected by synergistic effects of mutations in distant parts of the sequence, with respect to two single mutations acting independently. This might indicate that the most effective way to optimize proteins' solubility could involve the study of the synergistic effects of mutations, instead of implementing variants whose effect on solubility has been assessed in isolation.

In S2 Fig we show the mean predicted synergistic effects, averaged over the possible pairs of amino acids mutations. Residues such as A, N, P, W, Y and I are on average less prone to show synergistic effects, while C, D, Q, M and V are, on average, involved in stronger synergistic effects. Interestingly, among these residues, Cs appears to experience little influence from mutations of Qs, and Hs are generaly not influenced by mutations of Ps and Rs.

In Fig 5 we analysed the mean synergistic effects on the protein O26734 in function of the sequence separation $|i - j|$ between pairs of mutated residues at positions $i, j$. We see that residues which are very close ($1 \leq |i - j| \leq 10$) or very distant ($90 \leq |i - j| \leq 100$) tend to experience the highest synergistic effects (blue lines). In order to find an explanation to this behavior, we compared the magnitude of the synergistic effects with the distribution of the actual 3D distances between residues extracted from the 1JRM pdb structure. As shown in Fig 5, we noticed that a certain correlation exists between the magnitude of the synergistic effects and the mean contact distance between residues (red lines). The Pearson correlation between the mean predicted synergy and mean Angstrom distance between the residues' C-$\beta$ atoms is $r = 0.29$ (p-value = 0.003) and the Spearman's correlation is $r = 0.36$ (p-value = 0.0002). This shows that the attention-based architecture on which SKADE is built is indeed able to catch a

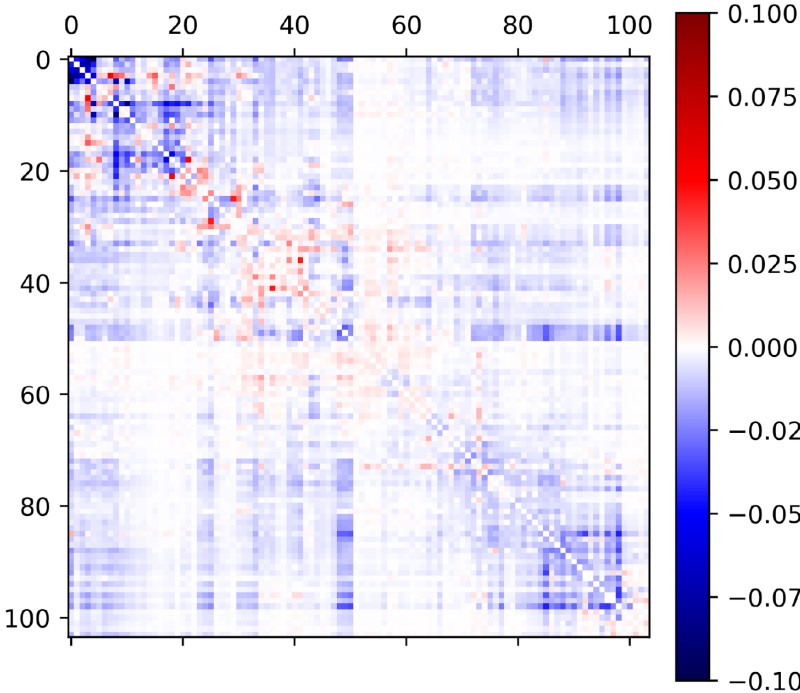

**Fig 4. Heatmap showing the average single versus synergistic effects $\Delta E = \Delta S_{single} - \Delta S_{pair}$ on each position of the O26734 protein.** Negative (blue) values indicate that synergistic effects are stronger, while positive (red) values indicate that the effect of independent mutations is higher.

glimpse of more complex structural aspects of proteins, such as the distribution of contacts. S8 Fig shows a scatter plot version of the same data.

## Discussion

In this study we propose a novel Neural Network (NN) architecture for the prediction of the solubility of protein sequences, and we exploit its attention-based interpretability to investigate the molecular forces driving protein solubility. This model, called SKADE, is based on a neural attention mechanism inspired from machine translation tasks and takes as input only the target protein sequence. SKADE's performance positively compares with state of the art solubility predictors, and the neural attention architecture offers some opportunities for the interpretation of the predictions, in a first step towards *opening* the Machine Learning (ML) *black-box*.

In this study, we indeed analyzed the attention profiles learned by the model during training to investigate whether they showed a significant correlation with biophysical properties of proteins that may relate to the solubility of the chain. Interestingly, we did not find any correlation with trivial biophysical characteristics of amino acids, such as described in biophysical propensity scales, but we showed that the *solubility profiles* extracted from the model can be used as an unsupervised predictor for aggregation-prone regions in proteins. This suggests that the attention-like mechanism in SKADE is indeed learning non-trivial biophysical emergent characteristics of the protein sequences and that uses them as building blocks to compute the final solubility prediction.

From the analysis of the attention profiles it also appears that the portions of the protein that carry the strongest signal when it comes to predict the protein solubility are the ones

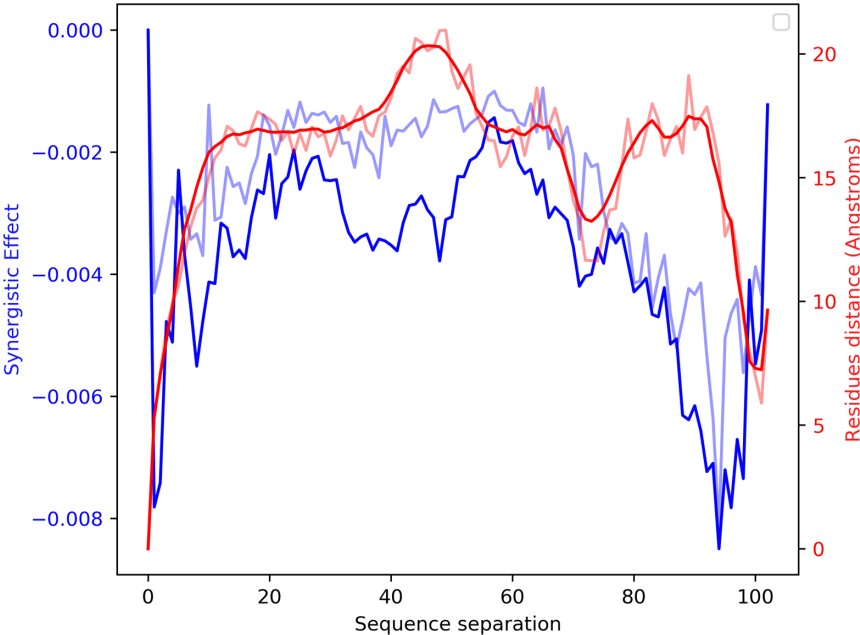

**Fig 5. Plot showing the correlation between the average spatial distance between residues at a certain sequence separation $|i - j|$ (red) with the magnitude of the synergistic effects between tandem mutations at the same positions $i, j$ (blue).** Solid colors indicate the mean, while light colors indicate the median values.

closer to the N- and C-termini, more specifically the first and last 20% of residues of the sequences.

Protein solubiliity predictors are generally used to determine which minimal set of mutations could increase the overall solubility of the protein, thus facilitating its expression and production. In our analysis we also show that, although SKADE has not been trained for the task, the model can distinguish between mutations that increase or decrease the protein solubility, and that our model can thus be used to perform in-silico mutational screening with the goal of optimizing the solubility of proteins by selecting an optimal set of mutations.

When solubility predictors are used to screen mutations to increase protein solubility, only *single* mutations are usually analysed, because the number of possible pairs (or triplets, quadruplets) of mutations grows exponentially. One of the advantages of SKADE is that its NN implementation can be heavily parallelized on GPU, and thus an unprecedented amount of predictions can be computed in very short time. SKADE can thus compute the in-silico mutational screening of pairs or triplets of mutations in minutes, thus including the possible synergistic effects of multiple mutations in this analysis. To show an example of this, we predicted all the possible pairs of mutations on the fairly short protein O26734, and we compared the solubility changes due to couples of independent mutations with respect to pairs of *tandem* mutations. From this analysis it appears that many regions of O26734 are predicted to be affected by synergistic interactions between mutations, and we thus hypothesize that modelling the synergistic effects of mutations may provide an optimal way towards the optimization of proteins with respect to specific biophysical desiderata.

Finally, while analysing the distribution of the magnitude of the synergistic effects with respect to the sequence separation, we noticed that the synergies predicted by SKADE from the O26734 protein sequence have a significant correlation with the average contact distance between residues, extracted from the corresponding PDB sequence. SKADE is thus able to

catch a glimpse of a complex emergent aspect of protein sequences, such as their contact density, from the sequence alone.

## Supporting information

**S1 Text. Supplementary information.** PDF file containing additional analyses.
(PDF)

**S2 Text. Supplementary information.** CSV file containing the aminoacid embedding values.
(CSV)

**S1 Fig. Plot showing the distributions of the $a_i \times p_i$ values over the protein sequence.** We grouped these values by using their relative position in their respective sequence. Although the distributions are quite similar, the variance is generally higher at the be- ginning and at the end of the sequences, indicating that the NN might find stronger signal in these regions.
(EPS)

**S2 Fig. Plot showing the average synergistic effect of pairs of mutations averaged over the type of mutated aminoacid.**
(EPS)

**S3 Fig. Plot showing the attention and prediction profiles of protein Q8TC59.**
(EPS)

**S4 Fig. Plot showing the attention and prediction profiles of protein Q9HBE1.**
(EPS)

**S5 Fig. Plot showing the attention and prediction profiles of protein P25984.**
(EPS)

**S6 Fig. Plot showing the 2 principal components of a PCA computed over the 20 dimensional embeddings learned by SKADE.**
(EPS)

**S7 Fig. Plot distributions of the mutations on the sequences in the CAMSOL dataset.**
(EPS)

**S8 Fig. Plot showing the correlation between the mean spatial distance (in Angstroms) and the average synergistic effects of pairs of residues at the same sequence separation in the O26734 protein.**
(EPS)

## Acknowledgments

DR is grateful to A. L. Mascagni for the constructive discussion and to A. Reynolds for the inspiration.

## Author Contributions

**Conceptualization:** Daniele Raimondi, Gabriele Orlando, Piero Fariselli, Yves Moreau.

**Data curation:** Daniele Raimondi, Piero Fariselli.

**Funding acquisition:** Daniele Raimondi, Yves Moreau.

**Methodology:** Daniele Raimondi, Gabriele Orlando.

**Software:** Daniele Raimondi.

**Validation:** Daniele Raimondi, Piero Fariselli.

**Writing – original draft:** Daniele Raimondi.

**Writing – review & editing:** Daniele Raimondi, Piero Fariselli, Yves Moreau.

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
