## [Decision Letter · Decision Letter 0]

7 Jan 2020

Dear Dr Raimondi,

Thank you very much for submitting your manuscript 'Insight into the protein solubility driving forces with neural attention' for review by PLOS Computational Biology. Your manuscript has been fully evaluated by the PLOS Computational Biology editorial team and in this case also by independent peer reviewers. The reviewers appreciated the attention to an important problem, but raised some substantial concerns about the manuscript as it currently stands. While your manuscript cannot be accepted in its present form, we are willing to consider a revised version in which the issues raised by the reviewers have been adequately addressed. We cannot, of course, promise publication at that time.

Sincerely,

Emil Alexov

Guest Editor

PLOS Computational Biology

Ruth Nussinov

Editor-in-Chief

PLOS Computational Biology

[LINK]

Reviewer's Responses to Questions

**Comments to the Authors:**

Reviewer #1: This is an interesting manuscript in the burgeoning field of protein solubility prediction. It is based on machine learning models, and produces results in line with other available methods. The authors describe their use of ML methods as novel in the field and this may be the case, I am afraid that I am not expert in ML methods. What I regard as novel are a couple of observations made in the manuscript, and that is where I would like to see more work carried out to clarify the results.

First, a statement is made to the effect that the predictions / model are not based on the commonly used biophysical properties (of amino acids). But, I do not see how this can be asserted if amino acids themselves are central to the prediction scheme - as I see it, sequence is the input to the training algorithm.

Second, and of the most interest to me, is the observation that N-t and C-t seem to possess (Fig. 2) more or less all the predictive power. I would not be surprised to find a tendency towards greater predictive power, but a flat line (Fig. 2) over the central 70% of the sequence I find amazing. Is it possible that the method is learning sequences (i.e. homology) and that this is best expressed at NT and CT when sequences of variable lengths are mapped to percentiles?

My own speculation is probably wrong, but I feel that the authors need more work here. e.g. What happens if the standard bioinformatics methods for reducing to non-redundant training set is used?

In the data reported for SKADE analysis of the CAMSOL mutation set, are all the mutations that affect solubility in the NT and CT regions - as might be expected given the results of Fig. 2?

Similarly, the synergistic mutations in later figures, are they also in NT and CT regions?

Reviewer #2: The paper presents a novel computational approach to predicting protein and peptide solubility from amino acid sequence alone SKADE. The model behind the method is a Recurrent Neural Network with soft self-attention and dot-product score function, built using Gated Recurrent Units. It is implemented in Python using Pytorch framework and supplies a pre-trained network for immediate applications.

SKADE is a great example of a neural network design without explicit feature engineering. The resulting model is comparable in performance or outperforming (depending on metrics) state of the art methods such as PaRSnIP and DeepSol that rely on knowledge-based and engineered features. The distributions of weights and scores along protein sequences suggest that N- and C-termini are most important determinants of protein and peptide solubility.

In addition, SKADE is able to identify mutations that increase or decrease the overall solubility of the protein, making it a useful and fast method for scanning mutagenesis, not limited to single mutations. Testing the effects of protein truncation on solubility is also possible with SKADE.

Major:

1. Although the model provides per-residue attention values and scores, interpretability of the methods remains limited. The only conclusion is that N- and C- termini are more important, but that is about it. Besides, these are the output layers of the network, what is inside is still a “black box” in terms of interpretability, I suggest deemphasizing this point in abstract and conclusions. The lack of correlations with biophysical properties contradicts this conclusion.

2. It is quite surprising to see that there is no correlation of attention and solubility profiles to any physico-chemical and/or structural properties of amino acids. Particularly, because these features work quite well in other solubility prediction methods. I think Table 1 in supporting information files does not comprehensively cover the biophysical features, for instance aggregation propensity, SASA. Besides, it is not clear how the correlations were calculated. Could the lack of correlation be related to scalability of attention/profile with sequence length? I.e. values of a short protein could be incomparable to values of a large protein.

3. According to neural network architecture it seems that the Embedding for converting 20 amino acid categories into a 20-dimensional tensor acts simply as a One Hot encoder. If there is any actual embedding that is learnt during training could you visualize the resulting amino acid similarities? Otherwise I suggest renaming embedding to encoding.

4. Python 2.7 is not acceptable by any means. This Python branch reached end of life on January 1 2020, is no longer supported. Moreover, is currently being removed from many computer systems for security concerns because it is unsupported. See https://devguide.python.org/#status-of-python-branches . It looks like there should not be any reasons not to update the code to Python 3.

5. Please format code according to PEP8 https://www.python.org/dev/peps/pep-0008/ Automatic conversion with autopep8 --aggressive could be used, for instance. Note that there is a lot of imported but unused modules, variables, and other issues that a lint program could fix, please also run pylint to get rid of the software issues.

Minor:

1. There is an imbalance between soluble and insoluble classes, since you have not used weighted sampling. To what extent do you think this imbalance affected the results? I suspect this could be the reason behind performance gap with respect to positive/negative class prediction separating SKADE and PaRSnIP.

2. The details of model convergence should be provided in supporting information. I could see that tensorboard was used in the program, so it should be possible to include those plots.

3. Line 98, why i<0 ?

4. Line 298: According to uniprot the protein should be 104, not 103 residues long https://www.uniprot.org/uniprot/O26734

5. The equation in line 300 is not clear, were the pairs of mutations generated including two mutations on the same position? Shouldn’t it be something like 104x19x19x103/2?

6. Current Pytorch v 1.3.1, version was 1.0.1 used in the paper. Model needs to be recalculated

7. Please submit the model to kipoi.org repository, that would make it much easier to run it

8. Peptide N- and C-termini are effectively charged residues in terms of their physico-chemical characteristics affecting solubility, i.e. Ala is hydrophobic, but will have a charge on the N-termini. However, it is not treated/encoded in any way different than any other amino acid. Was the reasoning behind it to an unbiased representation in the model or something else?

9. What is BAC in table 2?

10. Line 65, correlation is reported as AUC, which is not convenient

11. Line 319: “my” > “by”

12. In my opinion it is better to show dependence of synergetic effect on spatial distance directly as a scatter plot to emphasize on the correlation, particularly in proximity

13. In Figure 3 according to heatmap the first WT residue in O26734 should be V (the only white shaded residue), however this is not the case. What gives?

**Have all data underlying the figures and results presented in the manuscript been provided?**

Reviewer #1: Yes

Reviewer #2: Yes

PLOS authors have the option to publish the peer review history of their article (what does this mean?). If published, this will include your full peer review and any attached files.

Reviewer #1: No

Reviewer #2: Yes: Alexander Goncearenco

---

## [Decision Letter · Decision Letter 1]

10 Feb 2020

Dear Dr. Raimondi,

We are pleased to inform you that your manuscript 'Insight into the protein solubility driving forces with neural attention' has been provisionally accepted for publication in PLOS Computational Biology.

Before your manuscript can be formally accepted you will need to complete some formatting changes, which you will receive in a follow up email. A member of our team will be in touch within two working days with a set of requests.

Best regards,

Emil Alexov

Guest Editor

PLOS Computational Biology

Ruth Nussinov

Editor-in-Chief

PLOS Computational Biology

Reviewer's Responses to Questions

**Comments to the Authors:**

Reviewer #2: The authors responded to all my questions.

**Have all data underlying the figures and results presented in the manuscript been provided?**

Reviewer #2: Yes

PLOS authors have the option to publish the peer review history of their article (what does this mean?). If published, this will include your full peer review and any attached files.

Reviewer #2: Yes: Alexander Goncearenco

---

## [Editor Report · Acceptance letter]

6 Apr 2020

PCOMPBIOL-D-19-02103R1 

Insight into the protein solubility driving forces with neural attention

Dear Dr Raimondi,

I am pleased to inform you that your manuscript has been formally accepted for publication in PLOS Computational Biology. Your manuscript is now with our production department and you will be notified of the publication date in due course.

With kind regards,

Laura Mallard
